# DeepPheno: Predicting single gene loss-of-function phenotypes using an ontology-aware hierarchical classifier

**Maxat Kulmanov**, **Robert Hoehndorf** *

King Abdullah University of Science and Technology, Thuwal, Kingdom of Saudi Arabia

* robert.hoehndorf@kaust.edu.sa

**Data Availability Statement:** The authors confirm that all data underlying the findings are fully available without restriction. Source codes are available at https://github.com/bio-ontology-research-group/deeppheno. Required data files are

## Abstract

Predicting the phenotypes resulting from molecular perturbations is one of the key challenges in genetics. Both forward and reverse genetic screen are employed to identify the molecular mechanisms underlying phenotypes and disease, and these resulted in a large number of genotype–phenotype association being available for humans and model organisms. Combined with recent advances in machine learning, it may now be possible to predict human phenotypes resulting from particular molecular aberrations. We developed Deep-Pheno, a neural network based hierarchical multi-class multi-label classification method for predicting the phenotypes resulting from loss-of-function in single genes. DeepPheno uses the functional annotations with gene products to predict the phenotypes resulting from a loss-of-function; additionally, we employ a two-step procedure in which we predict these functions first and then predict phenotypes. Prediction of phenotypes is ontology-based and we propose a novel ontology-based classifier suitable for very large hierarchical classification tasks. These methods allow us to predict phenotypes associated with any known protein-coding gene. We evaluate our approach using evaluation metrics established by the CAFA challenge and compare with top performing CAFA2 methods as well as several state of the art phenotype prediction approaches, demonstrating the improvement of DeepPheno over established methods. Furthermore, we show that predictions generated by DeepPheno are applicable to predicting gene–disease associations based on comparing phenotypes, and that a large number of new predictions made by DeepPheno have recently been added as phenotype databases.

## Author summary

Gene–phenotype associations can help to understand the underlying mechanisms of many genetic diseases. However, experimental identification, often involving animal models, is time consuming and expensive. Computational methods that predict gene–phenotype associations can be used instead. We developed DeepPheno, a novel approach for predicting the phenotypes resulting from a loss of function of a single gene. We use gene functions and gene expression as information to prediction phenotypes. Our method

available for download at https://bio2vec.cbrc.kaust.edu.sa/data/deeppheno/.

**Funding:** This work was supported by funding from King Abdullah University of Science and Technology (KAUST) Office of Sponsored Research (OSR) under Award No. URF/1/3454-01-01, URF/1/3790-01-01, FCC/1/1976-28-01, FCC/1/1976-29-01. The funders had no role in study design, data collection and analysis, decision to publish, or preparation of the manuscript.

**Competing interests:** The authors have declared that no competing interests exist.

uses a neural network classifier that is able to account for hierarchical dependencies between phenotypes. We extensively evaluate our method and compare it with related approaches, and we show that DeepPheno results in better performance in several evaluations. Furthermore, we found that many of the new predictions made by our method have been added to phenotype association databases released one year later. Overall, DeepPheno simulates some aspects of human physiology and how molecular and physiological alterations lead to abnormal phenotypes.

This is a *PLOS Computational Biology* Methods paper.

## Introduction

Many human diseases have a genetic basis and are caused by abnormalities in the genome. Due to their high heterogeneity, many disorders are still undiagnosed and despite significant research their genetic basis has not yet been established [1]. Understanding how disease phenotypes evolve from an organism's genotype is a significant challenge.

Reverse genetic screens can be used to investigate the causality of perturbing molecular mechanisms on a genetic level and observing the resulting phenotypes [2]. For example, the International Mouse Phenotyping Consortium (IMPC) [3] aims to associate phenotypes with loss of function mutations using gene knockout experiments, and similar knockout experiments have been performed in several model organisms [4]. Further genotype–phenotype associations for the laboratory mouse and other model organisms are also systematically extracted from literature and recorded in model organism databases [5–8].

The similarity between observed phenotypes can be used to infer similarity between molecular mechanisms [9]. In humans, the Human Phenotype Ontology (HPO) provides a controlled vocabulary for characterizing phenotypes [10]. A wide range of genotype–phenotype associations have been created based on the HPO. Further information about genotype–phenotype associations is collected in databases such as Online Mendelian Inheritance in Man (OMIM) [11], Orphanet [12], ClinVar [13], and DECIPHER [14].

With the number of genotype–phenotype associations available now, it may be possible to predict the phenotypic consequences resulting from some changes on the level of the genotype using machine learning. Several methods have been developed to automatically predict or generate genotype–phenotype associations. To predict phenotype associations, these methods use different sources such as literature [15–17], functional annotations [15, 18, 19], protein–protein interactions (PPIs) [15, 20–22], expression profiles [23, 24], genetic variations [15, 25], or their combinations [15]. The general idea behind most of these methods is to find genetic similarities, or interactions, and transfer phenotypes between genes based on the assumption that similar or interacting genes are involved in similar or related phenotypes [26].

Phenotypes arise from complex biological processes which include genetic interactions, protein–protein interactions, physiological interactions, and interactions of an organism with environmental factors as well as lifestyle and response to chemicals such as drugs. A large number of these interactions can be described using the Gene Ontology (GO) [27], and GO annotations are available for a large number of proteins from thousands of species [28]. Further, in recent years, significant progress has been made in predicting the functions of uncharacterized proteins [29, 30].

We developed a hierarchical classification approach of predicting gene–phenotype associations from function annotations of gene products. Similarly to the HPO2GO [19] method, we use the GO function annotations as our main feature and predict HPO classes. We propagate both functional and phenotype annotations using the hierarchical structure (taxonomy) of GO and HPO and train a deep neural network model which learns to map sets of GO annotations to sets of HPO annotations. One limitation of predicting phenotypes from functions is that not all genes have experimental functional annotations. We overcome this limitation by using the function prediction method DeepGOPlus [31] which has been trained on a large number of proteins and can generate accurate functional annotations using only protein sequence. As DeepGOPlus is trained on a large number of proteins, including close and distant homologs of many human proteins, we hypothesize that it can also provide information about the physiological processes to which proteins contribute.

Phenotype prediction is a massively multi-class and multi-label problem, with an ontology as classification target. In DeepPheno, we implemented a novel hierarchical classification layer which encodes almost 4,000 HPO classes and their hierarchical dependencies in a single prediction model which we use during training and prediction. The novel classification layer is a more scalable and faster version of the hierarchical classification layer used in DeepGO [32], and allows us to encode a large number of taxonomic dependencies. Although it uses simple matrix multiplication and max-pooling operations to enforce hierarchical dependencies, to our knowledge, it is a first time that these kind of operations are being used to encode ontology structure while training and optimizing a neural network using backpropagation.

We evaluate DeepPheno using the latest phenotype annotations available from the HPO database [10] and using the evaluation dataset from the Computational Assessment of Function Annotation (CAFA) challenge [29], and we compare our results with the top performing methods in CAFA 2 [33] and phenotype prediction method such as HPO2GO [19], HTD/TPR [34] and PHENOstruct [15]. We demonstrate a significant improvements over the state of the art in each evaluation.

To further validate the usefulness of our phenotype predictions in computational biology, we test whether we can predict gene-disease associations from the predicted phenotype annotations. We compute semantic (phenotypic) similarity between gene–phenotype annotations and disease–phenotype annotations, and our results show that the phenotype annotations generated by DeepPheno are predictive of gene–disease associations; consequently, DeepPheno expand the scope of phenotype-based prediction methods, such as for gene–disease associations [35, 36] or used in variant prioritization [37–39], to all genes for which functions are known or can be predicted. We further analyzed the predictions generated by our method and found that, in average, more than 60% of the predicted genes for a phenotype interact with genes that are already associated with the phenotype, suggesting that some of our false positive predictions might actually be truly associated within a phenotype module. Finally, we show that many of the predictions made by DeepPheno in the past have recently been added to phenotype databases based on experimental or clinical evidence.

## Results

### DeepPheno predicts phenotypes associated with single gene loss of function

We developed a set of neural network model that predict phenotypes associated with genes. The models are trained using functions associated with gene products as input and loss-of-function phenotypes as output and relate aberrations in function to abnormal phenotypes that arise from these aberrations.

We evaluate our method on the phenotype annotations from the HPO database [10] released in June 2019. We randomly split the dataset into training, validation and testing sets; we split the data by genes so that if a gene is included in one of the three sets, the gene is present with all its annotations. We tune all parameters of our models using the validation set and report evaluation results on the unseen testing set. We train five neural network models using the same phenotype annotations and same training/testing split, but with different functional annotations as features. The first model is called DeepPhenoGO and it uses only experimental function annotations. The second model is called DeepPhenoIEA and is trained on all annotations from UniProtKB/SwissProt including predicted ones (i.e., including annotations with an evidence code indicated the annotation was electronically inferred). The predicted annotations are usually based on sequence or structural similarity and multiple sequence alignments. The third model is called DeepPhenoDG. We train DeepPhenoDG on functions predicted by the DeepGOPlus [31] method. The fourth model is called DeepPhenoAG which is trained on combined function annotations from UniProtKB/Swissprot and DeepGOPlus. Finally, the DeepPheno model was trained on all functional annotations and gene expression features (i.e., the anatomical locations at which genes are expressed). Phenotypes are structured based on the HPO, and we evaluate "flat" versions of each model that do not consider the ontology structure to determine the contribution of the hierarchical classification layers in DeepPheno. For flat predictions, we fix hierarchical dependencies by propagating all positive predictions using the *true path rule* [27] using the structure of the HPO ontology. In other words, we add all superclasses up to the root class to the predictions for each positively predicted class.

We first compare our results with phenotype annotations generated by the "naive" method (see Materials and methods for details). The naive method predicts the most frequently annotated phenotype classes in the training set for all genes in the testing set. It achieves an $F_{max}$ of 0.378 which is close to our results and higher than the state-of-the-art methods in the CAFA2 challenge. Despite such performance, naive annotations do not have any practical use as they are identical for every gene. Our neural network model achieves an $F_{max}$ of 0.437 when we train it with only experimental GO annotations and, as expected, it improves to 0.451 when we add predicted annotations in UniProtKB. We expected results to improve because experimental GO annotations are not complete and the electronically inferred function annotations can add missing information. The model trained with all annotations including DeepGOPlus, achieves the same $F_{max}$ of 0.451, but results in the best performance among all models in our $S_{min}$ (which also considers the specificity of the predicted phenotype classes, see Materials and methods) and AUPR evaluation. DeepPhenoDG model which is trained with only DeepGO-Plus annotations achieves $F_{max}$ of 0.444 which is slightly lower than the best result, but is higher than the model trained with experimental GO annotations. Adding gene expression features to functional annotations improve the model's performance further achieving highest $F_{max}$ of 0.457. In each case, the hierarchical classifiers improves the performance achieved by the flat classifier and *true path rule* propagation. In addition, we trained a Random Forest regression model DeepPhenoRF with all features used in DeepPheno. It resulted in a similar performance as DeepPheno in terms of $F_{max}$ and slightly better performance in recall and the $S_{min}$ evaluation. In all other evaluations our neural network based model resulted in better performance. Table 1 summarizes our results.

The results show that the DeepPheno model achieves the best performance when using both experimental and electronically inferred GO annotations and gene expression values, and achieves comparative performance by using only DeepGOPlus annotations which can be predicted from protein amino acid sequence information alone. We also show that the Random Forest model can achieve similar performance using DeepGOPlus annotations and expression values.

**Table 1. The comparison of 5-fold cross-validation evaluation performance on the June 2019 dataset and all HPO classes.**

| Method | $F_{max}$ | Precision | Recall | $S_{min}$ | AUPR | AUROC |
|---|---|---|---|---|---|---|
| Naive | 0.378 ± 0.005 | 0.355 ± 0.004 | 0.406 ± 0.007 | 123.546 ± 3.002 | 0.306 ± 0.006 | 0.500 ± 0.000 |
| DeepPhenoGOFlat | 0.431 ± 0.004 | 0.415 ± 0.007 | 0.450 ± 0.010 | 117.102 ± 2.627 | 0.412 ± 0.008 | 0.709 ± 0.017 |
| DeepPhenoGO | 0.437 ± 0.005 | 0.411 ± 0.007 | 0.467 ± 0.014 | 116.469 ± 2.440 | 0.421 ± 0.006 | 0.727 ± 0.010 |
| DeepPhenoIEAFlat | 0.448 ± 0.004 | 0.427 ± 0.006 | 0.471 ± 0.006 | 115.225 ± 2.773 | 0.434 ± 0.006 | 0.758 ± 0.004 |
| DeepPhenoIEA | 0.451 ± 0.004 | 0.434 ± 0.009 | 0.469 ± 0.008 | 115.294 ± 2.708 | 0.436 ± 0.006 | 0.761 ± 0.003 |
| DeepPhenoDGFlat | 0.442 ± 0.005 | 0.426 ± 0.006 | 0.459 ± 0.012 | 115.795 ± 2.395 | 0.427 ± 0.005 | 0.758 ± 0.004 |
| DeepPhenoDG | 0.444 ± 0.007 | 0.426 ± 0.013 | 0.463 ± 0.007 | 115.412 ± 2.296 | 0.431 ± 0.010 | 0.760 ± 0.005 |
| DeepPhenoAGFlat | 0.444 ± 0.006 | 0.422 ± 0.010 | 0.468 ± 0.006 | 115.802 ± 3.154 | 0.429 ± 0.008 | 0.752 ± 0.009 |
| DeepPhenoAG | 0.451 ± 0.004 | 0.428 ± 0.006 | **0.477 ± 0.009** | 114.894 ± 3.043 | 0.438 ± 0.005 | 0.764 ± 0.002 |
| DeepPhenoFlat | 0.451 ± 0.006 | 0.434 ± 0.015 | 0.471 ± 0.006 | 114.765 ± 2.558 | 0.437 ± 0.008 | 0.763 ± 0.010 |
| DeepPheno | **0.457 ± 0.007** | **0.445 ± 0.010** | 0.470 ± 0.009 | 114.045 ± 2.821 | **0.445 ± 0.009** | **0.766 ± 0.010** |
| DeepPhenoRF | 0.456 ± 0.007 | 0.437 ± 0.014 | **0.477 ± 0.015** | **112.152 ± 2.114** | 0.431 ± 0.008 | 0.733 ± 0.006 |

In addition, we report the 5-fold cross-validation evaluation performance for the 100 best-performing individual phenotype classes in S1 Table. We found that DeepPheno results in a good performance for several specific phenotypes. For example, the phenotype class *Elevated levels of phytanic acid* (HP:0010571) has $F_{max}$ of 0.960 or the class *Brushfield spots* (HP:0001088) has $F_{max}$ of 0.892. Both classes are specific classes and leaf nodes in the HPO. Furthermore, we computed the performance of HPO branches represented by a class and its subclasses that are predicted by DeepPheno. S2 Table shows the performance of the top 100 branches with at least 5 classes. Noticably, DeepPheno results in $F_{max}$ of 0.77 for *Mode of Inheritance* (HP:0000005) branch of HPO. Moreover, we used the information content measure (see Materials and methods) as a class specificity based on annotations and plotted the performance of each phenotype by class specificity. S1 Fig shows that there is no significant correlation between class specificity and performance.

To compare our method with other methods, we trained and tested our model using the CAFA2 challenge data, i.e., using the training and testing data as well as the ontologies provided in CAFA2 (see Materials and methods). We further evaluated annotations for CAFA2 targets provided by the HPO2GO method [19]. The top performing methods in CAFA2 achieve an $F_{max}$ of around 0.36 [33]. Our DeepPhenoGO model trained using only experimental GO annotations achieve $F_{max}$ of 0.379. Models which use predicted annotations improve $F_{max}$ score and we achieve the best $F_{max}$ of 0.398 with the DeepPhenoAG model trained with all GO annotations from UniProtKB/SwissProt and predicted by DeepGOPlus. To make a fair comparison we excluded gene expression features since they were not available at the CAFA2 challenge time. Fig 1 shows the comparison with CAFA2 top performing methods and HPO2GO, and Table 2 shows the performance results of DeepPheno on the CAFA2 benchmark set.

The PHENOstruct [15] and HTD/TPR [34] methods report the 5-fold cross-validation evaluation results on HPO annotations released on January 2014 (which is the CAFA2 challenge training data). We followed their experimental setup and evaluated our methods with 5-fold cross-validation. To tune and select the best prediction models, we used 10% of training data as validation set in each fold. Table 3 compares our results with the reported results [34] where authors separately evaluate predictions for three subontologies of HPO. On the *Organ (Phenotypic) abnormality* (HP:0000118) subontology, our method performs with $F_{max}$ of 0.42 which is the same for PHENOstruct and the second best result after TPR with SVM. On the *Mode of Inheritance* (HP:0000005) subontology, our method achieves $F_{max}$ of 0.72

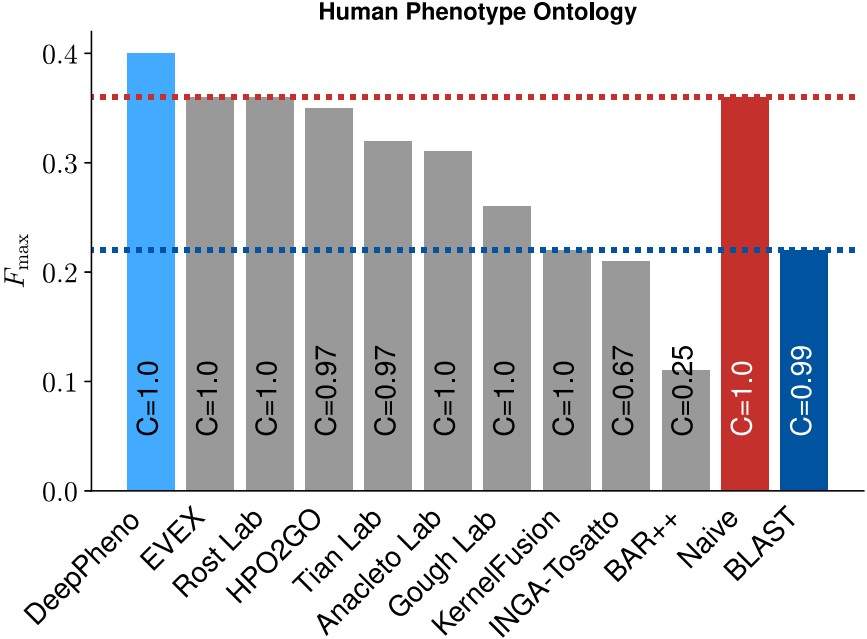

**Fig 1. Comparison of DeepPheno with CAFA2 top 10 methods and HPO2GO.**

which is slightly lower than PHENOstruct. Finally, our method outperforms all other methods in evaluation on the *Onset (Clinical modifier) subontology* (HP:0000004) with $F_{max}$ of 0.53. It is important to note that other methods rely on multiple sources of information such as PPIs, GO functions and text mining features, while DeepPheno uses only GO functions as features.

## Gene–disease associations using predicted phenotypes

Phenotype annotations have many applications, including prediction of candidate genes that are causally involved in diseases [36, 40–42]. For example, we can compare the similarity between gene–phenotype associations and disease phenotypes and prioritize candidate diseases for diagnosis [41], or prioritize genetic variants that are causative for a disease [37, 38]. These methods can suggest candidate genes for rare diseases and improve identification of causative variants in a clinical setting [41]. Here, our aim is to test whether the predicted phenotype annotations generated by DeepPheno are applicable for gene–disease association predictions.

**Table 2. The comparison of performance on the CAFA2 challenge benchmark dataset.**

| Method | $F_{max}$ | Precision | Recall | $S_{min}$ | AUPR | AUROC |
|---|---|---|---|---|---|---|
| Naive | 0.358 | 0.335 | 0.384 | 81.040 | 0.267 | 0.500 |
| DeepPhenoGO | 0.379 | 0.329 | 0.446 | 82.139 | 0.318 | 0.597 |
| DeepPhenoIEA | 0.396 | 0.379 | 0.415 | **79.461** | 0.348 | 0.645 |
| DeepPhenoDG | 0.392 | **0.388** | 0.397 | 81.369 | 0.339 | 0.621 |
| DeepPhenoRF | 0.391 | 0.378 | 0.405 | 80.545 | 0.338 | 0.619 |
| DeepPhenoAG | **0.398** | 0.340 | **0.482** | 79.616 | **0.350** | **0.648** |

**Table 3. The comparison of 5-fold cross validation performance on the CAFA2 challenge training dataset.**

| Method | AUROC | $F_{max}$ | Precision | Recall |
|---|---|---|---|---|
| Organ sub-ontology | | | | |
| TPR-W-RANKS | **0.89** | 0.40 | 0.34 | 0.48 |
| TPR-W-SVM | 0.77 | **0.44** | 0.38 | 0.51 |
| HTD-RANKS | 0.88 | 0.37 | 0.30 | 0.49 |
| HTD-SVM | 0.75 | 0.43 | 0.37 | 0.49 |
| PHENOstruct | 0.73 | 0.42 | 0.35 | **0.56** |
| Clus-HMC-Ens | 0.65 | 0.41 | 0.39 | 0.43 |
| PhenoPPIOrth | 0.52 | 0.20 | 0.27 | 0.15 |
| SSVM→Dis→HPO | 0.49 | 0.23 | 0.16 | 0.41 |
| RANKS | 0.87 | 0.30 | 0.23 | 0.43 |
| SVM | 0.74 | 0.42 | 0.36 | 0.50 |
| DeepPhenoAGFlat | 0.74 | 0.42 | 0.39 | 0.45 |
| DeepPhenoAG | 0.74 | 0.42 | **0.40** | 0.45 |
| Inheritance sub-ontology | | | | |
| TPR-W-RANKS | **0.91** | 0.57 | 0.45 | 0.80 |
| TPR-W-SVMs | 0.82 | 0.69 | 0.59 | 0.82 |
| HTD-RANKS | 0.90 | 0.57 | 0.44 | 0.81 |
| HTD-SVMs | 0.81 | 0.69 | 0.59 | 0.82 |
| PHENOstruct | 0.74 | **0.74** | **0.68** | 0.81 |
| Clus-HMC-Ens | 0.73 | 0.73 | 0.64 | **0.84** |
| PhenoPPIOrth | 0.55 | 0.12 | 0.16 | 0.10 |
| SSVM→Dis→HPO | 0.46 | 0.11 | 0.07 | 0.25 |
| RANKS | 0.90 | 0.56 | 0.43 | 0.81 |
| SVMs | 0.82 | 0.69 | 0.59 | 0.82 |
| DeepPhenoAGFlat | 0.67 | 0.72 | 0.65 | 0.80 |
| DeepPhenoAG | 0.68 | 0.72 | 0.64 | 0.81 |
| Onset sub-ontology | | | | |
| TPR-RANKS | **0.86** | 0.44 | 0.33 | **0.70** |
| TPR-SVMs | 0.75 | 0.48 | 0.38 | 0.66 |
| HTD-RANKS | **0.86** | 0.42 | 0.30 | 0.69 |
| HTD-SVMs | 0.74 | 0.46 | 0.37 | 0.67 |
| PHENOstruct | 0.64 | 0.39 | 0.31 | 0.52 |
| Clus-HMC-Ens | 0.58 | 0.35 | 0.27 | 0.48 |
| PhenoPPIOrth | 0.53 | 0.25 | 0.25 | 0.24 |
| SSVM→Dis→HPO HPO | 0.49 | 0.07 | 0.06 | 0.10 |
| RANKS | 0.83 | 0.41 | 0.30 | 0.67 |
| SVMs | 0.74 | 0.47 | 0.37 | 0.63 |
| DeepPhenoAGFlat | 0.64 | 0.52 | 0.46 | 0.62 |
| DeepPhenoAG | 0.66 | **0.53** | **0.47** | 0.60 |

One of the widely used methods for comparing ontology class annotations is semantic similarity [43]. We compute semantic similarity scores between gene–phenotype annotations and disease–phenotype annotations and use it to rank genes for each disease. Such an approach can be used to predict gene–disease associations based on phenotypes [36, 40] and allows us to determine how our predicted annotations can contribute to such an application.

As expected, phenotype annotations generated by the naive approach are not predictive of gene–disease associations and resulted in a performance with AUROC of 0.50. On the other

**Table 4. The comparison of 5-fold evaluation performance on gene–disease association prediction for the June 2019 dataset test genes.**

| Method | Hits@10 (%) | Hits@100 (%) | Mean Rank | AUROC |
|---|---|---|---|---|
| Naive | 1 | 9 | 522.72 | 0.50 |
| DeepPhenoGO | 10 | 35 | 307.36 | 0.70 |
| DeepPhenoIEA | 13 | 41 | 263.13 | 0.74 |
| DeepPhenoDG | 11 | 38 | 284.92 | 0.72 |
| DeepPhenoAG | 12 | 41 | 260.80 | 0.75 |
| DeepPheno | 12 | 41 | 260.05 | 0.75 |
| RealHPO | **90** | **96** | **22.03** | **0.98** |

hand, similarity scores from existing annotations from HPO performed nearly perfectly with a AUROC of 0.98. The reason for this nearly perfect prediction is that most of the OMIM diseases are associated with only one gene and share almost same phenotype annotations as their associated genes because of how the gene–phenotype associations have been generated [10]. Predicting gene–disease associations using the DeepPhenoDG models that rely on electronically inferred GO function annotations resulted in an AUROC of 0.72, which is slightly higher than predicting gene–disease associations using the model trained with experimental GO annotations which resulted in AUROC of 0.70. The best performance in AUROC of 0.75 among methods which use predicted phenotypes were achieved by our DeepPhenoAG and DeepPheno models. Table 4 summarizes the results. Overall, this evaluation shows that our model is predicting phenotype associations which provide a signal for predicting gene–disease associations and can be considered as features for this task.

## Evaluation of false positives

In our experiments, we consider the absence of knowledge about an association as a negative association. When we predict such an association, we consider this a false positive prediction. However, current gene–phenotype associations in databases are not complete and some of the false positive predictions generated by our method may actually be a correct association. To test this hypothesis, we evaluate our false positive predictions in three different ways. First, we check if our false positive annotations become true annotations in later releases of the HPO annotations database. Second, we evaluate if our predicted genes are interacting with a phenotype-associated gene using the underlying assumption that phenotypes are determined by network modules of interacting genes and gene products [44–46]. Finally, we investigate if some of our false positive predictions were reported in GWAS studies.

To evaluate our false positives, we used the phenotype annotations released by the HPO database in August 2020 and tested our model which was trained and tested on the 80/20% random split on a dataset from June 2019. We found that the proteins in the test set obtained 3, 207 new annotations in August 2020 and 898 (28%) of DeepPheno predictions that were counted as false in the June 2019 dataset appear in the August 2020 dataset. For example, NADH dehydrogenase, alpha 1 (*NDUFA1*) from the testing set of June 2019 has 13 phenotype annotations. In the August 2020 dataset, this gene collected 35 new annotations and 27 of them were predicted by our model. For example, DeepPheno predicted *Lactic acidosis* (`HP:0003128`) as a new phenotype associated with *NDUFA1* as well as several related phenotypes such as *Decreased liver function* (`HP:0001410`) and *Generalized hypotonia* (`HP:0001290`), as well as phenotypes such as *Microcephaly* (`HP:0000252`) and *Optic atrophy* (`HP:0000648`); all these phenotypes were added as new annotations of *NDUFA1* by August 2020. The predictions of DeepPheno were based on the GO function annotations for

*NDUFA1* as well as DeepGOPlus-predicted GO functions; the predicted functions include among others *NADH dehydrogenase (ubiquinone) activity* (GO:0008137) and *NADH dehydrogenase complex assembly* (GO:0010257) which are newly predicted by DeepGOPlus and not included in the experimental function annotations of *NDUFA1*. The NAD+/NADH ratio controls the activity of pyruvate dehydrogenase [47–49], and pyruvate dehydrogenase deficiency is one of the main causes of lactic acidosis [50, 51] as well as microcephaly and blindness [52]. The predictions made by DeepPheno indicate that our model was able to relate a decrease in metabolic function of NADH dehydrogenase to the downstream effects of NADH on pyruvate dehydrogenase, and then predict the phenotypes associated with it. One further indication for this mechanism is the prediction of DeepPheno of *Decreased activity of the pyruvate dehydrogenase complex* (HP:0002928) which had not been added as an annotation of *NDUFA1* in August 2020 and is still a false positive prediction by DeepPheno; however, this prediction indicates a mechanism for the majority of the phenotypes that were added by August 2020.

Similarly, the gene ceramide galactosyltransferase (*GALC*) had 27 phenotype annotations and obtained 125 new annotations in the HPO database of which 26 were predicted by DeepPheno. *GALC* is involved in Krabbe disease [53], a lysosomal storage disease with significant nervous system involvement [54] exhibiting a range of heterogeneous phenotypes, severity, and onset [55, 56]. While several major phenotypes of Krabbe disease (e.g., seizures, muscle weakness, blindness and optic atrophy, and hydrocephalus) were present in the 2019 dataset, broader and more specific phenotypes reflecting the phenotypic heterogeneity of Krabbe disease were added in 2020. Phenotypes correctly predicted by DeepPheno include *Ataxia* (HP:0001251) or *Dysarthria* (HP:0001260), which are specific neurological phenotypes also associated with Krabbe disease. These phenotypes are predicted from the asserted and predicted GO functions of *GALC* which include *galactosylceramide catabolic process* (GO:0006683), *myelination* (GO:0042552), and the cellular component *lysosome* (GO:0005764). Similarly to predictions for the *NDUGA1* gene, DeepPheno was able to identify the downstream effects of reduced galactosylceramide activity and correctly predict the phenotypes arising from them.

As further evaluation, we generated DeepPheno predictions for 18,860 genes with corresponding protein entries in Swissprot/UniProtKB. For every specific phenotype, we compared false positive genes and genes that are interacting with the phenotype-associated gene using the STRING database [57]. We then computed the percentage of overlap. We found that on average, 48% of our false positives interact with a phenotype-associated gene and may contribute to a phenotype-module within the interaction network. We tested whether this finding is significant by performing a random simulation experiment. For every phenotype, we generated the same number of random gene associations as our predictions and computed the average overlap of false positives and interacting genes. We repeated this procedure 1,000 times and tested the significance of having an average of 48% overlap. We found that random predictions give an average of 24% overlap and the probability of observing 48% overlap under a random assignment of genes to phenotypes is 0.02, demonstrating that the genes we predict for a phenotype are significantly more likely to directly interact with a known phenotype-associated gene.

In addition, we investigated in some cases whether the false positive gene–phenotype associations were reported to be significant in GWAS studies. We used the GWAS Catalog [58] to determine if some of our false predictions were reported in GWAS studies, and tested the phenotypes Type II Diabetes mellitus (HP:0005978) and Hypothyroidism (HP:0000821) since they are specific phenotypes in HPO and are very well studied with many GWAS

analyses. Type II Diabetes is associated with 136 genes in the HPO database and we predict 189 genes with DeepPheno. 69 genes from our predictions were already included in the HPO database and we consequently generated 120 false positives. We found that our false positive associations with genes WFS1, HNF4A, NPHP4, and TXNL4B were reported to be associated with Type II Diabetes in GWAS Catalog while the others were not. NPHP4 was also associated by a GWAS study using the UK Biobank dataset (https://www.nealelab.is/uk-biobank). For the Hypothyroidism (HP:0000821) phenotype, we predict 61 genes which are not in our training set. A GWAS study using the UK Biobank dataset reported an association with LGR4 gene which is in our false positive set. As we predict genes that interact with phenotype-associated genes, and while the predicted genes do mostly not reach genome-wide significance in GWAS studies, a possible explanation may be that some of the predicted genes have only low effect sizes, preventing them from being detected in a GWAS study. In future research, it may be possible to explore this hypothesis further by directly evaluating interaction models and Deep-Pheno's predictions on GWAS datasets.

## Discussion

DeepPheno can predict sets of gene–phenotype associations from gene functional annotations. Specifically, it is designed to predict phenotypes which arise from a loss of function (where functions are represented using the Gene Ontology) and we have illustrated how DeepPheno relates loss of functions to their downstream phenotypic effects. While DeepPheno was trained using phenotypes arising from the loss of function of a gene, its reliance on functions (instead of structural features) may allow it to also be applied to different alterations of gene function such as partial loss of function. Together with function prediction methods such as DeepGO-Plus [31], DeepPheno can, in principle, predict phenotype associations for protein-coding genes using only the protein's amino acid sequence. However, DeepGOPlus was trained on experimentally annotated sequences of many organisms, including several animal model organisms. It further combines global sequence similarity and a deep learning model which learns to recognize sequence motifs as well as some elements of protein structure. The combination of this information is implicitly used in DeepGOPlus and its predictions, and is therefore able to predict physiological functions that are closely related to the abnormal phenotypes predicted by DeepPheno.

## Evaluation

We evaluated DeepPheno on two datasets and compared its predictions with the top performing methods in the CAFA2 challenge. DeepPheno showed overall the best performance in the evaluation with time based split. However, when we compared the performance of DeepPheno on 5-fold cross-validation on CAFA2 challenge training set with other hierarchical classification methods such as PhenoStruct [15] and HTD/TPR [34], our method did not outperform HTD/TPR methods combined with support-vector machine classifiers and resulted in the same performance as PhenoStruct. We think that the main reason for this is that we only rely on function annotations and the other methods use additional features such as protein–protein interactions, literature and disease causing variants associated through gene-disease associations from HPO [10]. We did not use gene expression data because it was not available during CAFA2 challenge. However, in our experiment with recent data, we have shown that DeepPheno can easily combine features from multiple sources which resulted in improvement of its performance.

## Hierarchical classifier

We implemented a novel hierarchical classfication neural network in DeepPheno. It was inspired by our previous hierarchical classifer in DeepGO [32]. However, the version used in DeepPheno is significantly faster and scalable. The main difference here is that DeepPheno uses only one layer which stores ontology structure whereas DeepGO had a layer for each class in the ontology which required a connection to its children classes. Also, our new model achieves hierarchical consistency by a simple matrix multiplication operation followed by a MaxPooling layer and does not require complex operations. In DeepGO, the largest model can predict around 1, 000 classes while DeepPheno predicts around 4, 000.

We specifically compare DeepPheno with other hierarchical classification methods such as PhenoStruct [15] and HTD/TPR [34]. Also, we use the true path rule [27] to fix hierarchical dependencies of DeepPhenoFlat classifiers and compare them with our hierarchical classifiers. In all cases, the DeepPheno models outperform flat classifiers that apply the true path rule after predictions.

Hierarchical deep neural networks have also been used to simulate interactions between processes within a cell and predict (cellular) phenotypes, notably in the DCell model [59]. DCell established a correspondence between the components of a deep neural network and ontology classes, both to model the hierarchical organization of a cell and to provide a means to explain genotype–phenotype predictions by identifying which parts of the neural network (and therefore which cell components or functions) are active when a prediction is made. DeepPheno uses ontologies both as input and output and to ensure that predictions are consistent with the HPO, but does not directly enable the interpretability of models such as DCell. DeepPheno also solves a different problem compared to DCell; while DCell relates (yeast) genotypes to growth phenotypes, DeepPheno predicts the phenotypic consequences of a loss of function; while DCell can simulate the processes within a cell, DeepPheno aims to simulate some aspects of human physiology and the phenotypes resulting from altering physiological functions.

## Limitations and future research

Currently, DeepPheno suffers from several limitations. Firstly, we use mainly function annotations and gene expressions as features. This gives our model the ability to predict phenotypes for many genes; however, phenotypes do not only depend on functions of individual gene products but also they arise from complex genetic and environmental interactions. Including such information may further improve our model. Specifically, we plan to include different types of interactions between genes in order to improve prediction of complex phenotypes.

Secondly, DeepPheno currently can only predict a limited number of phenotypes for which we find at least 10 annotated genes. This limitation is caused by the need to train our neural network model and limits DeepPheno's ability to predict specific phenotypes which are the most informative. One way to overcome this limitation is to include phenotype associations with different evidence, such as those derived from GWAS study instead of using only phenotypes resulting from Mendelian disease as included in the HPO database.

Finally, DeepPheno uses a simple fully connected layer and sparse representation of functional annotations and do not considers the full set of axioms in GO and HPO. Although, this model gave us the best performance in our experiments, we think that more "complex" learning methods which encode all semantics in the ontologies need to be considered in the future.

## Conclusions

DeepPheno provides a fast and scalable way of generating phenotype annotations from functional annotations and expression values of genes. We developed a novel neural network based hierarchical classification layer which can be easily integrated into any neural network architecture and which is applicable for many prediction tasks with a large number of classes with hierarchical dependencies. Our model can combine different types of features (multi-modal); therefore, it can be extended further by adding new information. We have shown that the performance of the model increases with the number of training data and we believe that this will allow our model to improve further in future. Our predictions are useful in associating genes and diseases, and we make our predictions publicly available for most of human genes.

We observe that incorporating axioms in ontologies into optimization of learning task helps to improve the model's performance. Moreover, DeepPheno relies on ontologies to relate altered molecular functions and processes to their physiological consequences. This property of DeepPheno could be exploited for other applications that rely on deriving phenotypes from functional aberrations.

## Materials and methods

### Evaluation and training data

**Training and testing dataset.** We downloaded the Human Phenotype Ontology (HPO) version released on 15th of April, 2019 and phenotype annotations released on 3rd of June, 2019 from https://hpo.jax.org. The HPO provides annotations from OMIM and Orphanet. We use gene–phenotype annotations from both sources and build a dataset of 4,073 genes annotated with 8,693 different phenotypes. In total, the dataset has 529,475 annotations with an average of 130 annotations per gene after propagating annotations using the HPO structure. 3,944 of the genes map to manually reviewed and annotated proteins from UniProtKB/SwissProt [60]. We refer to this dataset as "June 2019". We use reviewed genes/proteins and split the dataset into 80% training and 20% testing sets. We use 10% of the training set as a validation set to tune the parameters of our prediction model. We generate 5 folds of random split and report 5-fold cross-validation results.

In order to have sufficient data for training, we select HPO classes that have 10 or more annotated genes. The model trained with this dataset can predict 3,783 classes. However, we use all 8,693 classes when we evaluate our performance counting the classes that we cannot predict as false negatives.

Our main features from which we predict phenotypes are gene functions. We use the Gene Ontology (GO) [27] released on 17th of April, 2019 and GO annotations (uniprot_sprot.dat. gz) from UniProtKB/SwissProt released in April, 2019. We construct three different datasets with different types of functional annotations for the same genes. In the first dataset, we use all annotations from the file which includes electronically inferred ones. The second dataset has only experimental functional annotations. We filter experimental annotations using the evidence codes `EXP`, `IDA`, `IPI`, `IMP`, `IGI`, `IEP`, `TAS`, `IC`, `HTP`, `HDA`, `HMP`, `HGI`, `HEP`. For the third dataset, we use GO functions predicted by DeepGOPlus [31]. The number of distinct GO classes which are used as features in our model is 24, 274.

Additionally, we perform an experiment where we combine gene functions with gene expressions data downloaded from Expression Atlas [61] Genotype-Tissue Expression (GTEx) Project (`E-MTAB-5214`) [62]. This data provides 53 features for each gene which are expression values of a gene for 53 different tissues. In total, there are expressions values for 40, 079 genes. We normalize values by genes and use zero for missing values.

**Comparison dataset.** To compare our method with state-of-the-art phenotype prediction methods we follow the CAFA [29] challenge rules and generate a dataset using a time based split. CAFA2 [33] challenge benchmark data was collected from January 2014 until September 2014. We train our model on phenotype annotations that were available before the challenge started and evaluate the model on annotations that appeared during the challenge period. Similarly, we use function annotations that were available before a challenge starts and train Deep-GOPlus on a UniProt/Swissprot version released in January 2014. We use the same versions of HPO and GO that were used in the challenge. Here, our model can predict 2, 029 classes that have 10 or more annotations for training.

In addition, phenotype prediction methods such as PHENOStruct [15] and HTD/TPR [34] report 5-fold cross-validation evaluation results using the phenotype annotations released in January 2014. We also follow their experimental setup and compare our results on this data.

**Protein–protein interactions data.** To evaluate false positive predictions generated by our approach we use protein–protein interactions (PPI) data. We download StringDB PPI networks [57] version 11 published in January, 2019. StringDB is a database of protein's functional associations which includes direct physical and indirect functional interactions. StringDB combines interactions from several primary PPI databases and adds PPI interactions that are predicted with computational methods. We use interactions with a score of 0.7 or above in order to filter high confidence interactions.

## Baseline and comparison methods

We use several approaches to benchmark and compare our prediction results. The "naive" approach was proposed by the CAFA [29] challenge as one of the basic methods to assign GO and HPO annotations. Here, each query gene or protein $g$ is annotated with the HPO classes with a prediction score computed as

$$S(g,p) = \frac{N_p}{N_{total}} \tag{1}$$

where $p$ is a HPO class, $N_p$ is a number of training genes annotated by HPO class $p$, and $N_{total}$ is a total number of training genes. It represents a prediction based only on the total number of genes associated with a class during training.

The CAFA2 [33] challenge evaluates several phenotype prediction methods which present state-of-the-art performance for this task. We train and test our model on the same data and compare our results with top performing methods. The CAFA3 [30] challenge also evaluated HPO predictions but did not release the evaluation results yet.

HPO2GO predicts HPO classes by learning association rules between HPO and GO classes based on their co-occurrence in annotations [19]. The idea is to map every HPO class $p$ to a GO class $f$ and score the mapping with the following formula:

$$S(p,f) = \frac{2 \cdot N_{p\&f}}{N_p + N_f} \tag{2}$$

where $N_{p\&f}$ is the number of genes annotated with both $p$ and $f$, $N_p$ is the number of genes annotated with $p$ and $N_f$ is the number of genes annotated with $f$. In the prediction phase the mappings are used to assign HPO classes to genes with available GO classes.

PHENOStruct [15] is a hierarchical multilabel classifier which predicts HPO classes using a structured SVM method. The PHENOStruct method relies on multiple gene features such as PPIs, GO functions, literature and genetic variants linked through gene-disease associations.

The Hierarchcal Top-Down (HTD) and True Path Rule (TPR) [34] methods are hierarchical ensemble methods which first train a flat classification method and update prediction scores using the ontology structure to fix inconsistencies of hierarchical label space. In the HTD strategy, scores are propagated from parent to children starting from the root class. If a parent has a lower score than its child then the child's score will be updated to the parent's score. In the TPR strategy, all superclasses of predicted classes are added to the predictions. Scores for the parent class are updated based on their children class scores. For example, a parent class can take an average value of positively predicted children or a maximum of all children scores. Both strategies guarantee consistency of prediction scores with hierarchical dependencies of the labels. The main limitation of these strategies is that they do not consider the hierarchical label space while training the models.

## Evaluation metrics

In order to evaluate our phenotype predictions and compare our method with other competing methods we use the CAFA [29] gene/protein-centric evaluation metrics $F_{max}$, area under the precision-recall curve (AUPR) and $S_{min}$ [63]. In addition to this, we report term-centric area under the receiver operating characteristic curve (AUROC) [64]. We compute AUROC for each ontology class and then take the average.

$F_{max}$ is a maximum gene/protein-centric F-measure computed over all prediction thresholds. First, we compute average precision and recall using the following formulas:

$$pr_i(t) = \frac{\sum_p I(f \in P_i(t) \land p \in T_i)}{\sum_p I(p \in P_i(t))} \tag{3}$$

$$rc_i(t) = \frac{\sum_p I(f \in P_i(t) \land p \in T_i)}{\sum_p I(p \in T_i)} \tag{4}$$

$$AvgPr(t) = \frac{1}{m(t)} \cdot \sum_{i=1}^{m(t)} pr_i(t) \tag{5}$$

$$AvgRc(t) = \frac{1}{n} \cdot \sum_{i=1}^{n} rc_i(t) \tag{6}$$

where $p$ is an HPO class, $T_i$ is a set of true annotations, $P_i(t)$ is a set of predicted annotations for a gene $i$ and threshold $t$, $m(t)$ is a number of proteins for which we predict at least one class, $n$ is a total number of proteins and $I$ is an identity function which returns 1 if the condition is true and 0 otherwise. Then, we compute the $F_{max}$ for prediction thresholds $t \in [0, 1]$ with a step size of 0.01. We count a class as a prediction if its prediction score is higher than $t$:

$$F_{max} = \max_t \left\{ \frac{2 \cdot AvgPr(t) \cdot AvgRc(t)}{AvgPr(t) + AvgRc(t)} \right\} \tag{7}$$

$S_{min}$ computes the semantic distance between real and predicted annotations based on information content of the classes. The information content $IC(c)$ is computed based on the annotation probability of the class $c$:

$$IC(c) = -\log(Pr(c|P(c))) \tag{8}$$

where $P(c)$ is a set of parent classes of the class $c$. The $S_{min}$ is computed using the following

formulas:

$$S_{\min} = \min_t \sqrt{ru(t)^2 + mi(t)^2} \tag{9}$$

where $ru(t)$ is the average remaining uncertainty and $mi(t)$ is average misinformation:

$$ru(t) = \frac{1}{n} \sum_{i=1}^{n} \sum_{c \in T_i - P_i(t)} IC(c) \tag{10}$$

$$mi(t) = \frac{1}{n} \sum_{i=1}^{n} \sum_{c \in P_i(t) - T_i} IC(c) \tag{11}$$

## Gene–disease association prediction

To evaluate predicted phenotype annotations we downloaded gene-disease association data from OMIM [65]. The OMIM database provides associations for around 6,000 diseases and 14,000 genes. We filter these associations with the genes from our randomly split test set and their associated diseases. In total, the dataset has 561 associations of 395 genes with 548 diseases.

We predict an association between gene and disease by comparing their phenotypes. Our hypothesis is that if a gene and disease are annotated with similar phenotypes then there could be an association between them [36]. We compute Resnik's [66] semantic similarity measure for pairs of phenotype classes and use the Best-Match-Average (BMA) [67] strategy to combine similarites for two sets of annotations. We use the similarity score to rank diseases for each gene and report recall at top 10 rank, recall at top 100, mean rank and the area under the receiver operating characteristic curve (AUROC) [64] for each prediction method.

Resnik's similarity measure is defined as the most informative common anchestor (MICA) of the compared classes in the ontology. First, we compute information content (IC) for every class with following formula:

$$IC(c) = -\log(p(c))$$

Then, we find Resnik's similarity by:

$$Sim_{Resnik}(c_1, c_2) = IC(MICA(c_1, c_2))$$

We compute all possible pairwise similarities of two annotation sets and combine them with:

$$Sim_{BMA}(A, B) = \frac{avg\left(\max_{c_2 \in B}(s(c_1, c_2))\right) + avg\left(\max_{c_2 \in A}(s(c_1, c_2))\right)}{2}$$

where $s(x, y) = Sim_{Resnik}(x, y)$.

## DeepPheno model

The aim of DeepPheno is to predict phenotypes which result from the loss of function of a single gene. In order to achieve this goal, we need to combine different types of features such as functions of gene products, molecular and multi-cellular interactions, pathways, and physiological interactions. Our approach is to use existing experimental and predicted function annotations of proteins, and learn the associations between combinations of functions and

phenotype annotations. We use OMIM [11, 65] diseases and their phenotype annotations from the HPO [10] to learn associations between sets of GO functional annotations and sets of HPO phenotype annotations. Since experimental function annotations are not available for all gene products, we also utilize the sequence-based function prediction method DeepGOPlus [31], which uses information of function annotations in many organisms, to fill this gap and predict the functions of gene products. Consequently, DeepPheno can predict phenotypes for all protein-coding genes with an available sequence.

Our phenotype prediction model is a fully-connected neural network followed by a novel hierarchical classification layer which encodes the ontology structure into the neural network. DeepPheno takes a sparse binary vector of functional annotation features and gene expression features as input and outputs phenotype annotation scores which are consistent with the hierarchical dependencies of the phenotypes in HPO; the input vector is deductively closed following the taxonomy and axioms in the GO ontology, i.e., function annotations are propagated. Fig 2 describes the model architecture. The first layer is responsible for reducing the dimensionality of our sparse function annotation input and expression values are concatenated to it. The second layer is a multi-class multi-label classification layer with sigmoid activation functions for each neuron. We use a dropout layer after the first fully-connected layer to avoid overfitting during training. The number of units in the second layer is the same as the number of prediction classes and its output is considered as the output of a "flat" classifier, i.e., a classifier which does not take the taxonomy of HPO into account. Our novel hierarchical classification layer computes the score for a class by selecting the maximum score of its descendants (i.e., subclasses according to the HPO axioms).

DeepPheno's hierarchical classification layer was inspired by the hierarchical classifier of DeepGO [32]. In DeepGO, each output class had a classification layer which was connected to its direct children's classifiers with a maximum merge layer. The main limitation of this method was its performance both in terms of training and runtime, since all the computations were performed sequentially. Consequently, the model did not scale to large ontologies such as GO or HPO. The novel hierarchical classification layer we implement in DeepPheno overcomes the limitations of the hierarchical classification previously used in DeepGO by using matrix multiplication operations and MaxPooling layers to encode the consistency constraints of phenotype annotations that arise from the ontology's taxonomic structure. We implement the layer by multiplying the sigmoid output vector with a binary vector encoded for the hierarchical dependencies in GO, where a value of "one" indicates that a class at this positions is a descendant of the class to be predicted; this multiplication is then followed by MaxPooling layer. Formally, given a vector $\mathbf{x}$ for a set of phenotype classes $P$ such that $x_i$ is a prediction score for a phenotype $p_i$ at position $i$, then we define the hierarchical classification function $h(\mathbf{x})$ as:

$$h(\mathbf{x}) = \max_{p_i \in P}(\mathbf{x} \cdot s(p_i)) \tag{12}$$

where $s$ is a function which returns a binary vector that encodes the subclass relations between $p_i$ and all other phenotype classes. The value of the vector $s(p_i)$ at position $j$ is 1 if $p_j$ is a (reflexive) subclass of $p_i$ ($p_j \sqsubseteq p_i$), and 0 otherwise. The function returns a vector of the same size as $\mathbf{x}$ which will be consistent with the *true-path-rule* because a prediction score of a class will be the maximum value of the prediction scores of all its subclasses. The use of elementwise matrix multiplication and pooling allows this hierarchical classification function to be implemented efficiently (in contrast to the sequential computation model of DeepGO) and we use this classification function during training of DeepPheno as well as during prediction. Fig 3 provides an example of the hierarchical classification function.

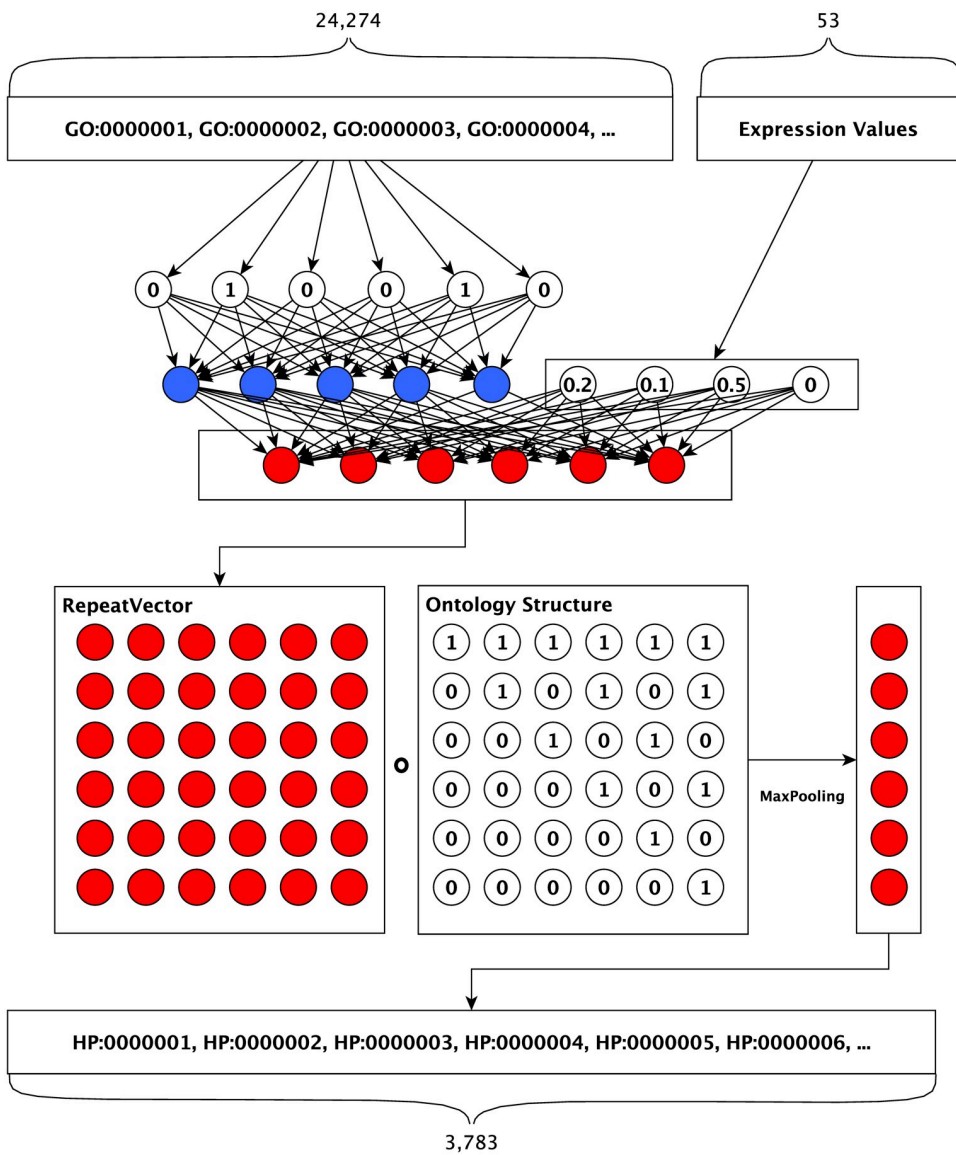

**Fig 2. Neural network model architecture.**

## Training and tuning of model parameters

We evaluated several models with two, three and four fully connected layer models. We selected the number of units for each layer from {250, 500, . . ., 4000} with dropout rate from {0.2, 0.5} and learning rate from {0.01, 0.001, 0.0001} for the Adam optimizer [68]. We trained the models with mini-batch size of 32. We performed 50 trials of random search for best parameters for each type of the models and selected the best model based on validation loss. We use the TensorFlow 2.0 [69] machine learning system with Keras API and tune our parameters with Keras Tuner.

Our model is trained and tuned in less than 1 hour on a single Nvidia Quadro P6000 GPU. In average, it annotates more than 100 samples per second.

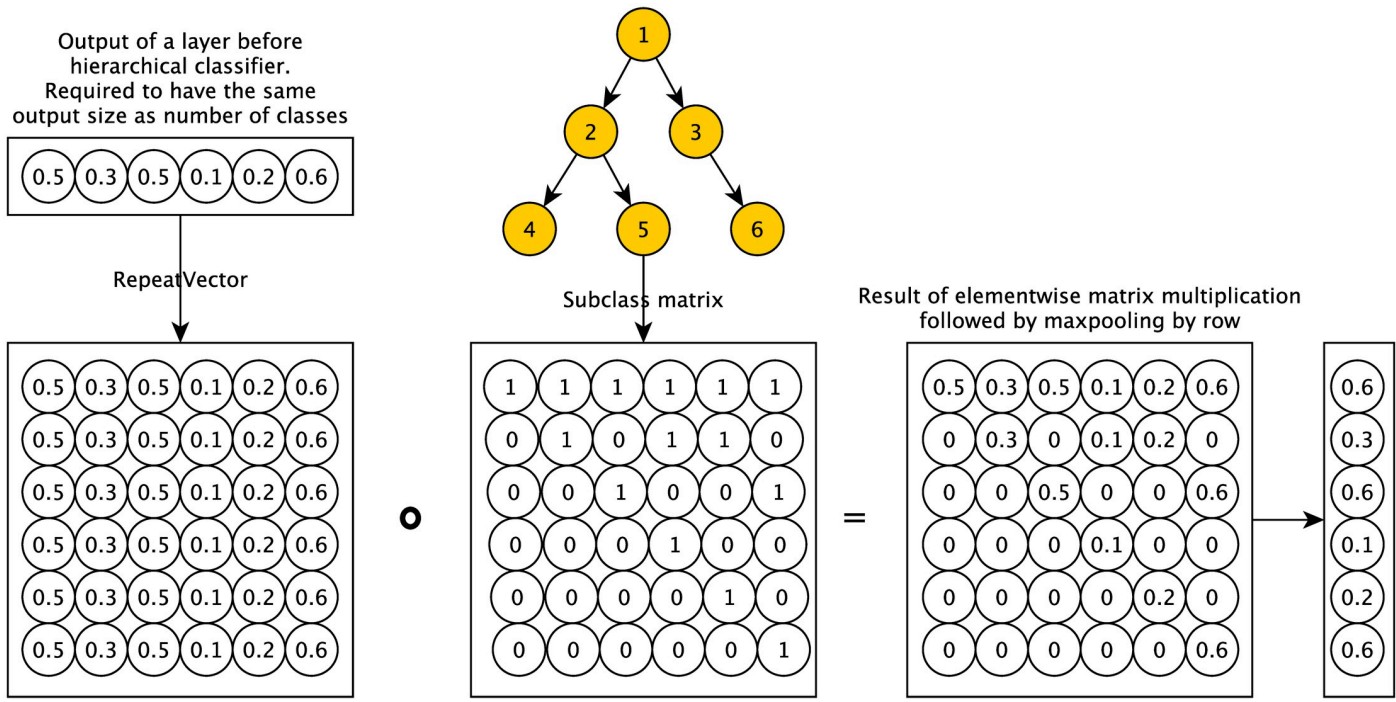

**Fig 3. Example of hierarchical classification layer operations.**

## Supporting information

**S1 Table. Performance of DeepPheno by phenotypes (Top 100).**
(PDF)

**S2 Table. HPO branches with significantly better performance (Top 100).**
(PDF)

**S1 Fig. Performance of each phenotype class by its specificity measure based on the number of annotations.**
(PDF)

## Acknowledgments

We acknowledge the use of computational resources from the KAUST Supercomputing Core Laboratory.

## Author Contributions

**Conceptualization:** Maxat Kulmanov.

**Formal analysis:** Maxat Kulmanov, Robert Hoehndorf.

**Funding acquisition:** Robert Hoehndorf.

**Investigation:** Maxat Kulmanov, Robert Hoehndorf.

**Methodology:** Maxat Kulmanov, Robert Hoehndorf.

**Project administration:** Robert Hoehndorf.

**Software:** Maxat Kulmanov.

**Supervision:** Robert Hoehndorf.

**Validation:** Maxat Kulmanov, Robert Hoehndorf.

**Writing – original draft:** Maxat Kulmanov, Robert Hoehndorf.

**Writing – review & editing:** Maxat Kulmanov, Robert Hoehndorf.

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
