## [Decision Letter · Decision Letter 0]

16 Aug 2020

Dear Mr. Kulmanov,

Thank you very much for submitting your manuscript "DeepPheno: Predicting single gene loss-of-function phenotypes using an ontology-aware hierarchical classifier" for consideration at PLOS Computational Biology.

As with all papers reviewed by the journal, your manuscript was reviewed by members of the editorial board and by several independent reviewers. In light of the reviews (below this email), we would like to invite the resubmission of a significantly-revised version that takes into account the reviewers' comments.

Reviewer 3 notes certain structural issues with the manuscript, and I agree with these. I nearly sent the manuscript back before review, but elected in the end to send it out for review with plans to note the structural considerations at this stage. As I was reading the manuscript, I also found myself thinking conceptually about the structure of the method proposed in Ma et al. (https://www.nature.com/articles/nmeth.4627). Discussing how this work fits conceptually with related work in the area (also noted by Reviewer 3) would make it more accessible to the interested reader who may not know the details of every method within the field.

We cannot make any decision about publication until we have seen the revised manuscript and your response to the reviewers' comments. Your revised manuscript is also likely to be sent to reviewers for further evaluation.

Sincerely,

Casey S. Greene

Guest Editor

PLOS Computational Biology

Jian Ma

Deputy Editor

PLOS Computational Biology

Reviewer 3 notes certain structural issues with the manuscript, and I agree with these. I nearly sent the manuscript back before review, but elected in the end to send it out for review with plans to note the structural considerations at this stage. As I was reading the manuscript, I also found myself thinking conceptually about the structure of the method proposed in Ma et al. (https://www.nature.com/articles/nmeth.4627). Discussing how this work fits conceptually with related work in the area (also noted by Reviewer 3) would make it more accessible to the interested reader who may not know the details of every method within the field.

Reviewer's Responses to Questions

**Comments to the Authors:**

**Reviewer #1: **The authors are predicting genotype-phenotype association using a neural network where GO annotations of gene functions are used as features. The model is trained and tested on Human Phenotype Ontology (HPO) data and CAFA2 data. In terms of evaluation metrics, the authors are by and large using the metrics used in the CAFA2 project.

On the CAFA2 benchmark dataset DeepPheno has the highest F_max score compared to all the top methods in CAFA2. This shows the model is indeed learning to connect genotype with phenotype. On the CAFA2 training data sub-ontologies, the performance is mixed where in many cases other methods are performing better. Overall, the paper is increasing the state-of-the-art performance in many aspects for the genotype-phenotype association problem which is praiseworthy.

Having said that, I have the following comments about some aspects-

1. I would like to compare the performance on CAFA2 datasets with a vanilla random forest that is taking the same input as DeepPheno as my first baseline. Random forests often surprisingly provide a strong baseline that is comparable or better than neural networks.

2. I am not convinced about the novelty of hierarchical classifier the authors are claiming. I don’t think this needs to be emphasized on the paper. Authors can provide their justification but this is a simple matrix multiplication.

3. I am not convinced of the analysis of the false positive predictions the authors performed. I don’t think the analysis done with the STRING dataset brings any confirmation or refutation of the false positives. The anecdotal examples mentioned in the False positive section also show that the majority of false positive predictions do not correspond to genotype-phenotype associations. The extrapolation by the authors seems a bit far fetched.

4. There is no guideline on the github page of the paper on how to apply their method on new datasets. Other researchers should be able to apply this method on their dataset with minimum knowledge.

Overall, this is a straight-forward paper that improves upon the performance by current methods. I would make some of the other claims in a more cautious manner.

**Reviewer #2:** Reproducibility Report has been uploaded as an attachment.

**Reviewer #3:** In this manuscript, the authors proposed a new gene-loss-of-function based phenotype prediction method that utilizes HPO terms, entitled DeepPheno, based on machine/deep learning. The study is relevant as novel in silico approaches are always required in the fields of protein function prediction and disease association prediction. The manuscript is mostly written well in terms of the use of English; but the sectioning of the text is a little bit problematic. The proposed methodology is novel and effective; however, there are critical issues related to the current state of the manuscript. Below, I list my specific concerns/issues:

Major issues:

1) The compartmentalization of the information into sections and sub-sections has issues, which makes it difficult to read the manuscript. For example, some technicalities regarding the structure of DeepPheno is provided at the beginning of the Results section, instead of the Methods section (i.e., the sub-section entitled DeepPheno Model should be under the methods). Additionally, some of the small-scale analyses are given in the discussion section (i.e., under False Positives sub-section), instead of the Results section. Sectioning should be re-considered from beginning to the end of the manuscript.

2) An additional test should be done considering the performance of DeepPheno on phenotypic terms from varying specificity groups. This grouping may either be based on the levels on the hierarchy of HPO or the number of annotated genes. Since DeepPheno uses deep neural networks and require a certain number of training instances to be able to predict a term, this analysis and its discussion may be critical and will help the reader in terms of machine learning algorithm selection according to the task and data at hand, in future predictive modeling studies.

3) There should be a use-case study to indicate the real-world value of DeepPheno, beyond performance calculations over the known data. This can be a case where DeepPheno predicts a specific gene-HPO association, which was not presented in the latest version of HPO, thus counted as a false positive. This case can be investigated over the literature to find an evidence for the predicted relation, as a means of validation. There already are a few examples in the discussion part regarding type II diabetes and GWAS; however, this example is also based on statistics and do not delve into the biological relevance of why those 4-5 genes should be associated with type II diabetes.

4) There are two points regarding a related and cited study, HPO2GO. First of all, the idea of relating the occurrence of a phenotype to the lost function, and thus, using the GO annotations of genes/proteins to predict their phenotypic implications has already been proposed in the HPO2GO study. The same logic is used for DeepPheno as well. This inspiration should be mentioned in this manuscript by referring to HPO2GO in the relevant parts of the text, at least along with the other works that inspired this idea.

Second, considering Table 2 and Figure 2:

In its own paper, HPO2GO’s finalized CAFA2 performance is reported to be Fmax = 0.35. Here in thew proposed manuscript, the reported performance of HPO2GO is similar to one of its sub-optimal versions explained in the corresponding article. Since CAFA challenge clearly states the train/test datasets to be used, and directly provides the performance calculation scripts, the results reported in different papers (where the authors state that they obey the CAFA rules) are directly comparable to each other. As a result, it is better to use the optimal CAFA2 results reported in the HPO2GO paper.

5) The information about the neural network layering is confusing. In figure 1 and the related text, it is stated that there are 2 layers: “The first layer is responsible for reducing the dimensionality of our sparse function annotation input and expression values are concatenated to it. The second layer is a multi-class multi-label classification layer with sigmoid activation functions for each neuron.” However, in the Training and tuning of model parameters sub-section it is stated that the authors evaluated several models with two, three and four fully connected layer models. Where are these models and their performance results? What are the details of these 3-4 layered models? Did the authors settle for the 2 layered model at the end? If so, what was the reason behind? Was it that the 2 layered model performed the best? Or, did the performance gain by increasing the layers did not compensate for the increase in computational complexity?

Related to this, a few hyper-parameters have been mentioned under Training and tuning of model parameters sub-section. How about the rest of the widely considered hyper-parameters in deep learning-based method development studies (e.g., using mini batches or not, if so, what is the size; and did you use batch normalization), have these been optimized as well, or the default value has been incorporated directly?

6) "Our phenotype prediction model is a fully-connected neural network followed by a novel hierarchical classification layer which encodes the ontology structure into the neural network. "

The technical details of this hierarchical classification layer are explained; however, this is a critical step in the proposed method, and the authors emphasize this as one of the important values added to the literature by this study. As a result, it should be explained and discussed in more detail. It is not clear how the hierarchical structure of the ontologies is taken into account inside the deep neural network. The authors stated that, this is achieved by multiplying the values of neurons at the second layer, with a matrix that represents the ontology structure, and then, max pooling is applied. Is this done to transfer, for example, the positive prediction given to a child term to its parent term, so that, even if the parent term did not receive a prediction itself, the system will change this to a positive prediction because its descendant received a positive prediction? If so, this will always behave towards increasing the number of predictions, and thus, the false positives. Moreover, are the flat versions the same models, only without this operation? Please provide precision and recall values in Table 1 along with the given metrics so that the reader can observe the effect of this operation and its advantages/disadvantages over the flat predictions. Please explain this hierarchical operation in more detail over a toy-example (e.g., imaginary prediction of an HPO term with and without the hierarchical processing).

7) The sizes of the feature vectors should be provided (in terms of both GO and gene expression).

8) “The best performance in AUROC of 0.75 among methods which use predicted phenotypes were achieved by our DeepPhenoAG and DeepPheno models. Table 4 summarizes the results. Overall, this evaluation shows that our model is predicting phenotype associations which can be used for predicting gene-disease associations.”

It is not clear how did authors draw this conclusion. Although higher than Naïve, AUROC values between 0.70-0.75 is not considered sufficiently high, so that they could be used with confidence. You can state that this is better compared to random prediction by a margin, but this value alone is not sufficient to say that DeepPheno can be used for predicting gene-disease associations, without further analysis.

9) “Specifically, it is designed to predict phenotypes which arise from a complete loss of function of a gene.”

Authors emphasize this at multiple locations of the manuscript; however, it’s not clear why this should be a complete loss of function. It can very well be a partial loss of function that give rise to the occurrence of a phenotype. Authors should either explain this in detail or change their statement.

Minor issues:

1) “Similarity between observed phenotypes can be used to infer similarity between

molecular mechanisms, even across different species. In humans, the Human Phenotype Ontology (HPO) [8] provides an ontology for characterizing phenotypes and a wide range of genotype{phenotype associations have been created based on the human phenotype ontology.”

Please re-write these sentences, they are problematic in terms of the use of language.

2) “We use 10% of the training set as a validation set to tune the parameters of our prediction model. We generate 5 folds of random split and report 5-fold cross-validation results.”

It is not clear how this is done. Normally, when a 5-fold CV is carried out it means that the training dataset is split into 5 pieces and each fold comprise 20% of the dataset. What is the meaning behind using 10% of the training set as a validation set here?

3) “We use interactions with a score of 700 or above in order to filter high confidence interactions.”

Scores in StringDB vary between 0 and 1, should this be 0.7?

4) “The value of the vector si at position j is 1 iff pj is a (reflexive) …”

Typo.

**Have all data underlying the figures and results presented in the manuscript been provided?**

Reviewer #1: Yes

Reviewer #2: None

Reviewer #3: Yes

PLOS authors have the option to publish the peer review history of their article (what does this mean?). If published, this will include your full peer review and any attached files.

Reviewer #1: No

Reviewer #2: **Yes: **Anand K. Rampadarath

Reviewer #3: No
---

## [Decision Letter · Decision Letter 1]

20 Oct 2020

Dear Mr. Kulmanov,

We are pleased to inform you that your manuscript 'DeepPheno: Predicting single gene loss-of-function phenotypes using an ontology-aware hierarchical classifier' has been provisionally accepted for publication in PLOS Computational Biology.

Best regards,

Casey S. Greene

Guest Editor

PLOS Computational Biology

Jian Ma

Deputy Editor

PLOS Computational Biology

Your manuscript has now been re-reviewed and is acceptable for publication. I would note that the software that you are describing is currently distributed through GitHub, which is not archival in a way that is ideal for scholarly work. I have requested an archive at Software Heritage, which is now available: https://archive.softwareheritage.org/browse/origin/directory/?origin_url=https://github.com/bio-ontology-research-group/deeppheno . If you would like to have a directly citable, archived version of your software then I would strongly encourage you to upload the software to figshare( https://figshare.com/ ) and to provide the figshare DOI in a software availability section of your manuscript. Please also note the reproducibility report. The results could not be reproduced due to dependency issues with TensorFlow. It could be wise to have a group independently test your software and instructions to improve the instructions that you provide to readers.

Reviewer's Responses to Questions

**Comments to the Authors:**

Reviewer #1: Thank you for addressing my concerns.

Reviewer #2: Reproducibility report has been uploaded as an attachment.

Reviewer #3: I would like to thank authors for properly addressing my concerns.

**Have all data underlying the figures and results presented in the manuscript been provided?**

Reviewer #1: Yes

Reviewer #2: None

Reviewer #3: Yes

PLOS authors have the option to publish the peer review history of their article (what does this mean?). If published, this will include your full peer review and any attached files.

Reviewer #1: No

Reviewer #2: No

Reviewer #3: **Yes: **Tunca Dogan

---

## [Editor Report · Acceptance letter]

12 Nov 2020

PCOMPBIOL-D-20-01270R1 

DeepPheno: Predicting single gene loss-of-function phenotypes using an ontology-aware hierarchical classifier

Dear Dr Kulmanov,

I am pleased to inform you that your manuscript has been formally accepted for publication in PLOS Computational Biology. Your manuscript is now with our production department and you will be notified of the publication date in due course.

With kind regards,

Nicola Davies
